# Fine Alignment of Thermographic Images for Robotic Inspection of Parts with Complex Geometries

**DOI:** 10.3390/s22166267

**Published:** 2022-08-20

**Authors:** Carmelo Mineo, Nicola Montinaro, Mario Fustaino, Antonio Pantano, Donatella Cerniglia

**Affiliations:** 1Institute for High-Performance Computing and Networking, National Research Council of Italy, 90146 Palermo, Italy; 2Department of Engineering, University of Palermo, 90128 Palermo, Italy

**Keywords:** robotics, thermography, non-destructive testing, image alignment, image blending

## Abstract

Increasing the efficiency of the quality control phase in industrial production lines through automation is a rapidly growing trend. In non-destructive testing, active thermography techniques are known for their suitability to allow rapid non-contact and full-field inspections. The robotic manipulation of the thermographic instrumentation enables the possibility of performing inspections of large components with complex geometries by collecting multiple thermographic images from optimal positions. The robotisation of the thermographic inspection is highly desirable to improve assessment speed and repeatability without compromising inspection accuracy. Although integrating a robotic setup for thermographic data capture is not challenging, the application of robotic thermography has not grown significantly to date due to the absence of a suitable approach for merging multiple thermographic images into a single presentation. Indeed, such an approach must guarantee accurate alignment and consistent pixel blending, which is crucial to facilitate defect detection and sizing. In this work, an innovative inspection platform was conceptualised and implemented, consisting of a pulsed thermography setup, a six-axis robotic manipulator and an algorithm for image alignment, correction and blending. The performance of the inspection platform is tested on a convex-shaped specimen with artificial defects, which highlights the potential of the new combined approach. This work bridges a technology gap, making thermographic inspections more deployable in industrial environments. The proposed fine image alignment approach can find applicability beyond thermographic non-destructive testing.

## 1. Introduction

Non-destructive Testing (NDT) comprises highly multidisciplinary groups of analysis techniques used throughout science and industry to evaluate materials’ properties and ensure the integrity of components/structures without causing damage to them [1]. In civil and industrial manufacturing, the increasing deployment of smart/composite materials demands high integrity and traceability of NDT measurements, combined with rapid data throughput. Traditional manual inspection approaches are insufficient in some scenarios since they produce a manufacturing process bottleneck [2]. Therefore, there are fundamental motivations for increasing automation in NDT. Computer-Aided Design (CAD) has been extensively used in engineering design phases. Computer-Aided Manufacturing (CAM) also allows large components to be produced quickly through combinations of traditional subtractive approaches and novel additive manufacturing processes [3]. As a result, large components with complex geometries have become very common in modern structures. NDT inspection is still often performed manually by technicians who typically must move appropriate probes over the contour of the part surface. Manual scanning requires trained technicians and results in a prolonged inspection process for large samples. Automation of NDT is required to cope with the inspection of such structures. Robotic manipulation of NDT sensors also plays an essential role in inspecting parts made of composite materials. A fundamental issue with composite components is that parts designed as identical can have significant deviations from the CAD model. Composite parts suffer from inherent but different part-to-part springiness out of the mould, which presents a significant challenge for precision NDT measurement deployment. While manual scanning may remain a valid approach for some specific areas of a structure, developing reliable automated solutions has become an industry priority to drive down inspection times and costs. An industrial robot is an automatically controlled, reprogrammable, multipurpose manipulator programmable in three or more axes [4]. Many manufacturers of industrial robots have produced robotic manipulators with excellent positional accuracy and repeatability. In the spectrum of robot manipulators, some modern robots have suitable attributes to develop automated NDT systems. They present precise mechanics, the possibility to accurately master each joint and the ability to export positional data at high update rates. The key challenges to face when developing a robotic NDT system include integrating the NDT instrumentation with the robotic manipulator, creating a suitable robot inspection path for the part under inspection, and developing software for NDT data collection and visualisation. These challenges have been addressed by several applications of six-axis robotic arms for the inspection of parts through automated ultrasonic techniques [5,6,7]. Robotic ultrasonic inspection has become commonplace thanks to the research investments driven by the aerospace sector in the suitability of ultrasonic techniques to inspect critical aerospace components. Some works have presented robotic ultrasonic inspection systems capable of achieving high data throughputs, accompanied by bespoke software for data visualisation and analysis [5,8]. Automated geometry mapping has also been demonstrated using robotically manipulated metrology sensors [9].

Besides these techniques, other types of inspections have not reached the same level of robotisation; this is the case for thermographic testing, also known as thermal imaging, infrared (IR) thermography or simply thermography. It is an NDT imaging technique that allows the visualisation of heat temporal patterns in an object or a scene and is based on the principle that two dissimilar materials possessing different thermophysical properties produce two distinctive thermal signatures that can be revealed by an infrared sensor, such as an IR thermal camera [10,11,12]. Although a thermographic setup in reflection mode, with a heat excitation source and an IR camera on the same side of the part under inspection, is not well suited to detect defects located deep in the volume of a component, it presents some advantages over ultrasonic-based inspections. It is contactless and full-field, meaning that the whole area of a component detectable within the field of view of an IR camera is inspected remotely at once. Schmidt and Dutta [13] proposed using industrial robots as manipulators to perform active thermography in 2012. The robotic manipulation of the thermographic instrumentation can enable the possibility of performing inspections of large components with complex geometries by collecting multiple thermographic images at given positions. Despite preliminary investigations [13,14], the robotisation of the thermographic inspection method has not been fully exploited to date due to the lack of a suitable approach capable of aligning automatically-collected thermographic images. The importance of consistent registration of NDT data in CAD models is highlighted in [15]. Aligning thermographic images for NDT analysis is not trivial since accurate and consistent pixel blending must be guaranteed and is crucial to facilitate defect detection and sizing. Inaccurate alignment and blending may create unreal artefacts in the composite thermography image and cause false-positive flaw detection. In this work, an innovative inspection platform was conceptualised and implemented, consisting of a pulsed thermography setup, a six-axis robotic manipulator and a novel algorithm for image transformation, alignment and blending. The performance of the inspection platform is tested on a convex-shaped specimen with artificial defects, highlighting the potential of the new combined approach.

The remaining part of this work is organised as follows. Section 2 reviews the theoretical principles of thermographic inspection and provides scientific literature references. Following a detailed clarification of the origin of the misalignment in robotically-acquired images and the limitations of existing image alignment algorithms, Section 3 describes the novel image alignment and blending algorithm developed by this work. Section 4 introduces the automatic thermography setup used to validate the proposed method. Section 5 presents the experimental results. The outcomes of this work and the method’s performance and prospects are discussed in Section 6.

## 2. Thermography Principles

Thermography, as introduced above, can be deployed through different techniques [16]. The essential equipment for manual (not automated) thermography includes an IR camera, a computer to record (and sometimes process) data and a monitor to display images. The main classification of the thermographic techniques differentiates between passive and active techniques. Passive thermography exploits the fact that materials and structures may naturally be at different (higher or lower) temperatures than the background. For example, the human body is generally at a higher temperature than the ambient; hence it is easily detected by an IR camera without additional stimulation. Conversely, an external stimulus is needed in active thermography to produce a thermal contrast in the object’s surface. Active techniques are particularly suited to non-destructive testing since an object containing internal defects (such as voids, delaminations and/or inclusions of foreign material) will require the excitation of thermal disequilibrium to produce a distinctive surface thermal signature detectable with an IR camera. In the realm of active thermographic techniques, pulsed thermography (PT) has broad applicability in NDT. When an object’s surface is heated through a short (a few milliseconds) energy pulse of light radiation, a series of thermal waves with different amplitudes and frequencies propagate inside the object medium in a transient mode. The surface temperature is monitored under the principle that defective areas cool down (or heat up) at a different rate than non-defective areas [17,18,19]. It is known that the thermal wave originating from the energy pulse can be decomposed into a multitude of individual sinusoidal components and that it is possible to link temporal and frequency domains. In pulsed phase thermography (PPT), the PT is combined with the phase and frequency concepts of lock-in thermography (LT), where specimens are subject to a periodical excitation [12,20,21,22]. Flash lamps generate a heat pulse of high intensity and low duration. The subsequent temperature decay is then acquired over a truncation window.

Once raw data are collected, there are multiple techniques to analyse the data. One approach consists of calculating the Discrete Fourier Transform (DFT) to evaluate the thermal response’s frequency content. The phase of specific harmonic content can finally be obtained and presented as a phasegram, an image where the scalar value associated with each pixel represents the phase. Any discontinuity in phase contrast is either caused by the object geometry or indicates a potential flaw. In the PPT approach, whereas deeper anomalies are expected to be better contrasted in low-frequency phasegrams, high-frequency phasegrams probe better for superficial issues. The signal normalisation inherent in evaluating the phase is also expected to reduce the counter effects of non-uniform heat deposition and environmental reflections [23]. It must be noted that the terms phasegram(s) and thermographic image(s) are used as synonyms in the remainder of this paper.

## 3. Fusion of Multiple Thermographic Images

### 3.1. Misalignment Issue in Robotically-Acquired Images

Robotic NDT inspections generally occur in a well-structured environment, where the part position is precisely registered with respect to the robot reference system. Great care is dedicated to ensuring the robot tool path is accurately referenced to the sample reference frame to ensure effective data collection during automated inspections [24]. Despite the efforts, a deviation between the actual tool path and the ideal tool path always remains due to the following reasons: (i) the physical tolerances in the robot joints; (ii) the geometric deviations in the mounting support of the sensing instrumentation; (iii) the residual inaccuracy in the calibration of the part position; (iv) the deviation between the actual sample geometry from the part digital counterpart. For these reasons, the resultant data usually reveal some imperfect alignment when they are encoded through robot positional feedback and plotted in the form of a single map. For robotic thermographic inspections, the problem translates to evident misalignment of the thermographic images. The issue may be mitigated through an external metrology tracking system (e.g., a six-DoF laser tracker), capable of measuring the position of the sensing instrumentation with respect to an absolute reference frame. However, such metrology systems are expensive and can increase the overall complexity of robotic inspection systems. In robotic machine vision systems, the exact position of the camera with respect to the robot mounting point is calibrated through the hand-eye calibration method [25], which is based on the knowledge of the camera’s intrinsic parameters, such as focal length, aperture, field-of-view and resolution, and on the capture of a calibration pattern (e.g., a checkerboard) from different viewpoints. However, this method is not always applicable to thermographic cameras since they do not usually have a visible-light imaging sensor (RGB sensor). A similar method based on calibration patterns with different thermal infrared emissivity could be adopted to calibrate IR cameras [26]. This work developed a practical solution consisting of correcting each image’s plotting location and its prospective aberrations to obtain a misalignment-free full-field view of the inspected sample. The remainder of this section explains the limitations of available image-stitching algorithms and the theoretical foundations of the proposed method herein.

### 3.2. Limitations of Existing Alignment Methods

Algorithms for aligning images and stitching them into seamless photo-mosaics are among the oldest and most widely used in computer vision. The alignment of images requires establishing mathematical relationships that map pixel coordinates from the unaligned images to their aligned versions. Five parametric 2D planar transformations have been defined [27] (see Figure 1). Each one of these transformations can be described by a transformation matrix $\left(\tau \left(p\right)\right)$, with p being a vector of parameters. Pure translation can be written as $x^{\prime }=x+t$ or $x^{\prime }=\tau \left(p\right)\cdot x=\left[\begin{array}{cc} I & t \end{array}\right]\cdot x$, where $x=\left\{x,\;y,\;1\right\}$ is the vector of coordinates of the untransformed image pixel and $x^{\prime }$ denotes the coordinates of the same pixel in the transformed image, I is the (2 × 2) identity matrix, and $t={\left[\begin{array}{cc} t_{x} & t_{y} \end{array}\right]}^{\prime }$ is the translation vector, containing two parameters (respectively, the translation along the *x*-axis and the translation along the *y*-axis). The Euclidean transformation is written as $x^{\prime }=\tau \left(p\right)\cdot x=\left[\begin{array}{cc} R & t \end{array}\right]\cdot x$, where R is the 2D rotation matrix. Thus, Euclidean transformation depends on three parameters: $t_{x}$, $t_{y}$, and an angle θ (for the rotation matrix). Euclidean distances are preserved. The similarity transformation, also known as scaled rotation, preserves angles between lines. It is expressed as $x^{\prime }=\left[\begin{array}{cc} sR & t \end{array}\right]\cdot x$, where s is the scale parameter that brings the parameter counter to four. It must be noted that s is a scalar and the scaling operation is intended to be isotropic. The affine transform is written as $x^{\prime }=\tau \left(p\right)\cdot x=A\cdot x$, where A is an arbitrary 2 × 3 matrix with six parameters. Parallel lines remain parallel under affine transformations. Projective transformation, also known as perspective or homography, is expressed as $x^{\prime }=\tau \left(p\right)\cdot x=H\cdot x$, where H is an arbitrary 3 × 3 matrix:(1)$$x^{\prime }=\left[\begin{array}{ccc} h_{00} & h_{01} & h_{02}\\ h_{10} & h_{11} & h_{12}\\ h_{20} & h_{21} & 1 \end{array}\right]\cdot x$$

Thus, perspective transformation requires eight parameters and preserves straight lines.

Assuming the choice of a suitable motion model to transform each image, a typical strategy to align a collection of images consists of aligning the images in pairs. In order to align a pair of images, it is necessary to devise some methods to estimate the parameters to apply the selected transformation to one image while the other is kept fixed. One approach is to shift or warp the first image relative to the other and measure how much the pixels agree. The first methods to quantitatively measure such agreement are often called “direct methods”, based on pixel-to-pixel matching [29]. These methods are usually slow since the number of pixel pairs to evaluate can be very large. Direct methods work by directly minimising pixel-to-pixel dissimilarities; a different class of algorithms works by extracting a sparse set of features and then matching these to each other [27,30,31]. Feature-based approaches have the advantage of being more robust against scene movement, are potentially faster, and can be used to automatically discover the adjacency (overlap) relationships among an unordered set of images [32].

Although feature-based approaches work well to create panoramas of scenes with enough distinguishable features, they are not suited to align multiple images for NDT applications. Non-destructive testing aims to detect defects in parts and/or structures. As such, besides the presence of intrinsic geometrical details (e.g., borders and corners), most images may appear relatively featureless since the presence of defects is not the norm. An attempt to use a feature-based alignment approach was presented in [33], where the authors note the need to mark artificial points on the background of a test objective to obtain the mapping matrix from two-dimensional (2D) thermal wave imaging data to the 3D spatial coordinate’s digital model. On the other hand, feature-based approaches can also fail if plenty of spatially periodic features are present in the images, which can be the case for industrial components due to stiffeners/stringers, heat dissipators and/or fixturing holes. Direct methods are less prone to failure caused by a lack of image features or abundance of periodicity since they can leverage any consistent low-contrast gradient to find the optimum image transformation parameters. However, the scientific literature does not show any solution readily available to work with the scalar information present in each pixel of thermographic images. As stated above, thermographic images differ from RGB or grayscale images since the pixel values may represent phases (expressed in degrees or radians) and may be negative values. Moreover, the optimum solution to align and stitch multiple thermographic images can not progress pairwise. Although it can work only for images taken in a single row, like in the case of a horizontal panorama, robotic thermographic inspection generally collects images through a raster tool-path, with multiple images arranged in multiple passes. A pairwise image-stitching algorithm would produce a visible drift between adjacent passes due to the progressive summation of alignment errors.

### 3.3. Fine Pixel-Based Alignment Method

This work developed a direct method capable of simultaneously aligning multiple images. The method is suitable to be used when the rough position of the camera (the shooting pose of each image) is known. That is the case for robotically acquired images, where the camera position is obtained from the robot’s positional feedback. Given a set of images, knowledge of camera shooting poses allows skipping the search for the adjacency relationships among the set. Knowing the scale factor makes it possible to convert the pixel index coordinates to real-world coordinates and identify the overlap between the images. The scale factor can easily be calculated by measuring the size of a known object or the known distance between two points in an image in terms of the number of pixels and considering the actual length it represents. Therefore, the algorithm herein is specifically targeted to perform a fine alignment of all images in the set. It is referred to as the Fine Pixel-based Alignment Method (FiPAM). It must be noted that FiPAM is currently suitable for aligning multiple mosaic images of a sample surface that curves only in one direction. Although the constraint of single direction curvature is a significant limitation, it does not impede using FiPAM for mosaic images of any surface belonging to the large family of cylindrical surfaces intended as “generalised cylindrical surfaces” [34]. Under that condition, all collected images can be transposed to a planar domain. Indeed, any cylindrical surface can be represented in the plane by “unrolling” it on a flat surface. An additional assumption is that the part surface captured within the camera field of view is sufficiently close to a flat plane. In other words, the ratio between the local surface curvature and the camera field of view must be small. Figure 2 illustrates a set of nine images used to explain the theoretical foundations of FiPAM.

Direct methods find the optimum alignment between a pair of images by an iterative search, where one image is transformed with respect to the other through one of the five planar transformations. To use a direct method, a suitable error metric must first be chosen to measure the goodness of the alignment. Given two images, with one image ($I_{0}\left(x\right)$) taken as a reference image sampled at discrete pixel locations ($x_{k}=\left\{x_{k},\;y_{k},1\right\}$), with k being the pixel index, we wish to find the optimum transformation parameters that align it with the second image ($I_{1}\left(x\right)$), which is kept fixed. The error metric is defined as the sum of squared differences (SSD) of the pixel values of $I_{1}$ at the transformed pixel locations and the reference values of $I_{0}$. This kind of function has been successfully used in the image processing literature, with different aims (e.g., inpainting [35]). Given a transformation ($\tau \left(p\right)$), with p being a vector of parameters, we have:(2)$$SSD\left(\tau \left(p\right)\right)=\underset{k}{\sum }{\left[I_{1}\left(\tau \left(p\right)\cdot x_{k}\right)-I_{0}\left(x_{k}\right)\right]}^{2}$$

The optimum set of parameters ($p^{*}$) can be found by solving a least-squares problem of this SSD function. Since the transformation allows multiple degrees of freedom (DoFs) for the image, this is a multi-parameter problem. Therefore, a suitable search technique must be devised. The most straightforward technique would be to exhaustively try all possible alignments (full search). In practice, this would be too slow and is not practicable. Several works have developed hierarchical coarse-to-fine search techniques based on image pyramids [27] when the approximate image alignment is unknown. In this work, since the approximate position of each image is assumed to come from the known camera pose, it has been decided to limit the search space by setting lower and upper bounds for the transformation parameters.

Regarding the set of images in Figure 2 and Figure 3 illustrates all the overlaps between image #4 and its neighbour images. Given a positive scalar value herein named “offset” (o), it is possible to draw shrunk overlap areas whose boundary is at distance o from the boundary of the original overlap areas. The actual value to use for o depends on the expected maximum entity of misalignment caused by the inaccuracy in robotic manipulation of the camera and by the deviations in the physical camera support. Assuming these offset areas move with image #4 and the original overlap areas stick with the parent neighbour image, the bounds of the transformation parameters guarantee that the offset overlap areas remain within the original overlap footprints. Generalising Equation (2) to allow simultaneous alignment of multiple images, FiPAM is based on the following SSD function.
(3)$$SSD\left(p\right)=\overset{n}{\underset{i=1}{\sum }}\overset{n}{\underset{j=1}{\sum }}\underset{k}{\sum }{\left[I_{j}\left(\tau_{i}\left(p\right)\cdot x_{k}\right)-I_{i}\left(\tau_{j}\left(p\right)\cdot x_{k}\right)\right]}^{2}\;\;\;\;\;\;\mathrm{with}\;\left(j\neq i\right)\;\mathrm{and}\;\left(k\in K_{i,j}\right).$$

$K_{i,j}$ is the set of pixel indices that fall within the offset area, produced by the overlap between the *i*th and the *j*th image. Assuming a set of n images, Equation (3) is the sum of squared differences of the pixel intensity values of the *j*th image and the ith image. Crucially, the overlap pixel locations of the *j*th image are transformed according to the transformation matrix of the *i*th image ($\tau_{i}\left(p\right)$) and the locations of the *i*th image are transformed according to the transformation matrix of the *j*th image ($\tau_{j}\left(p\right)$).

Now, it must be noted that the vector p includes all the parameters required in the transformation matrices, and only a subset of it is used to compute a single transformation matrix ($\tau_{i}\left(p\right),\;\mathrm{with}\;i=1:n$). Moreover, the summation is not evaluated for j=i (an image is always aligned with itself) and for combinations of i and j corresponding to images that do not overlap, where $K_{i,j}$ is an empty set. This formulation solves a typical problem with pixel-based methods, which is the possibility that parts of $I_{i}$ may lie outside the boundaries of $I_{j}$. This advantage follows directly from the constraints applied to the search space for the transformation parameters. Another aspect to discuss relates to the fact that the transformed pixel indices can be fractional, so a suitable interpolation function must be applied to evaluate the image intensities ($I_{i}$ and $I_{j}$). This work employs bi-cubic interpolation, which yields better results than bilinear interpolants [36]. It must be noted that Equation (3) does not require the image pixel values to be in a specific format. Thus, it can work with the phase values of thermographic phasegrams and images with three RGB colour channels, although it is also possible to first transform the images into a different colour space.

The mathematical parametric formulation of all transformation matrices pictured in Figure 1 was implemented in FiPAM. The formulation allows maximum flexibility in choosing the most suitable transformation for each image, meaning that all images in a set can be aligned using the same type of planar transformation, or each image can use a transformation of a different type. In other words, each image can be transformed by allowing different DoFs, which relate to a different number of parameters. Automating the selection of the optimum transformation for each image is out of the scope of this work. In practical situations, similarity or affine transformations produce satisfactory results if the part surface captured within the camera field of view is sufficiently close to a flat plane. Once the optimum transformation parameters are found, the aligned version of the *i*th image is computed by transforming its original discrete pixel locations with the following equation:(4)$$x_{i}^{\prime }=\tau_{i}\left(p^{*}\right)\cdot x_{i}$$

### 3.4. Image Blending

Aligning all images in a dataset is not sufficient to merge the images into a single composite image. Indeed, multiple aligned images may present significant differences in pixel intensities in overlapping areas. For RGB images, exposure differences are typically caused by ambient light changes during image capture. In active thermographic imaging, the same problem may be caused by the progressive increase of an object’s surface temperature when it is subject to multiple heat pulses. Image blending is usually accomplished through averaging the intensity of homologue/overlapping pixels or by using more sophisticated methods, such as “Laplacian pyramid blending” [37] and “Gradient-domain blending” [38]. Although these blending methods work well and have been implemented in many variants for consumer imaging (e.g., for panoramic image stitching), they cannot directly be used to blend images originating from NDT inspections. Indeed, in NDT images, it is necessary to retain the robustness of quantitative information (e.g., to perform pixel intensity comparisons) and avoid introducing any image processing artefacts. A typical challenge lies in removing low-frequency exposure variations while retaining sharp intensity gradients that may indicate the presence of small defects. In other words, it is necessary to prevent blurring. In this work, image blending has been solved through a method that preserves the valuable NDT information in each image. All pixel intensities in an image are offset by a unique value to maintain gradients unaltered. To explain this approach, Figure 4a,b provides an example of nine aligned images. The intensity discontinuity between any two overlapping images has been purposely emphasised. These example images do not contain high contrast features, which are typical for NDT images taken of a not-defected sample.

The idea is to shift the intensity of all pixels in an image vertically by a particular corrective value. Thus, n being the number of images in the set, it is necessary to compute a vector of  n scalar optimum intensity correction values ($c^{*}=\left[c_{1}^{*},c_{2}^{*},\;\ldots \;c_{i}^{*},\ldots c_{n}^{*}\right]$) that simultaneously correct all images in the set. These values may be positive or negative to produce an increase or a decrease in image pixel intensities. Interestingly, this computation can be formalised again through a least-squares problem of the following SSD function:(5)$$SSD\left(c\right)=\overset{n}{\underset{i=1}{\sum }}\overset{n}{\underset{j=1}{\sum }}\underset{h}{\sum }{\left\{\left[I_{j}\left(x_{h}\right)+c_{j}\right]-\left[I_{i}\left(x_{h}\right)+c_{i}\right]\right\}}^{2}\;\;\;\;\;\;\mathrm{with}\;\left(j\neq i\right)\;\mathrm{and}\;\left(h\in H_{i,j}\right),$$
where $H_{i,j}$ is the set of pixel indices that fall within the overlap between the aligned *i*th and *j*th image. It must be noted that the formulation of this SSD function follows the same approach used for the computation of the alignment parameters. The summation is not evaluated for j=i (no intensity self-correction is required) and for combinations of i and j corresponding to images that do not overlap, where $H_{i,j}$ is an empty set. Since intensity correction is performed after the alignment stage, Equation (5) does not perform any image transformation. Moreover, since the problem is limited to the computation of only one scalar parameter per image, convergence to a solution for Equation (5) is obtained faster than for Equation (3). Once the optimum intensity correction values are found, the matrix of corrected pixel intensities for the *i*th image (${\tilde{I}}_{i}$) is computed with the following equation.
(6)$${\tilde{I}}_{i}=I_{i}+c_{i}^{*}$$

Once all images are aligned and their intensity is corrected, the final composite image is obtained by applying the Laplacian pyramid blending, which allows a smooth transition between images. The application of blending at the end of the procedure is admissible since it does not introduce any image artefact when pixel intensity differences are low, which is the case after the phase of image pixel correction.

## 4. Robotic Thermography Setup

### 4.1. Inspection System Integration

Figure 5 illustrates the automatic thermography setup used in this work. The robotic manipulator was a KUKA KR10 R1100-2 arm [39], with a maximum payload of 11.1 kg and a maximum reach of 1101 mm. The setup was designed to perform PPT inspection in reflection mode, meaning that the flash lamp and the IR camera were always kept on the same side of the part under inspection. A custom-built supporting bracket was used to mount the flash lamp and the IR camera onto the robot and keep them in a fixed relative position during the inspection. The support allowed adjusting the orientation of the flash lamp to set the angular offset between the flash lamp illumination axis and the camera axis. This adjustment is not active because an actuator does not vary it during the execution of a robotic inspection path. However, keeping the camera focal distance constant for all data collection poses in a path makes it possible to manually set the optimum angular offset for any chosen camera focal distance before executing the inspection path. The heat source was an Elinchrom Twin X4 Lamphead EL20181, capable of releasing a pulse of 4800 W/s with a duration of 5.56 ms (1/180 s), powered by two power supplies in a parallel configuration [40]. The excitation source features a lightweight aluminium chassis, two twin flash tubes and twin cables connected to two Elinchrom 2400 RX power packs. The presence of two flash tubes and two power packs allows shorter flash durations and faster recycle times than a single flash tube connected to a single power pack, which is advantageous for the robotisation of the thermographic inspection. The IR camera was a cooled FLIR X6540sc IR-camera [41], equipped with a 50 mm F/2.0 lens; it has an adjustable acquisition rate of up to 125 Hz at full frame. The camera detector consists of 640 × 512 pixels, cooled by a Stirling thermodynamic cycle that uses an Indium-Antimonide fluid. The camera was connected to the computer through the Gigabit Ethernet link for full bandwidth data acquisition. The FLIR ResearchIR Max^®^ software (version 4.40.1), running on the computer, enabled the initial configuration of the camera and the reception of the thermographic data during the robotic inspection. DFT was used to evaluate the frequency content of the thermal response.

### 4.2. Sample

The sample was an epoxy specimen reproducing the curved geometry of a compressor blade. The specimen was produced by pouring a mix of liquid epoxy resin and a hardener into a mould. The resultant polymerised sample had one convex side, one concave side, and a varying thickness. The curvature of both surfaces is constrained to one direction. Six flat bottom holes (FBHs), three with square sections and three with round sections, were machined on the concave side of the sample as artificial defects. Thus, the FBHs are not visible from the convex side of the sample. Figure 6a,b illustrate the sample geometry, its main dimensions, and the position and size of the FBHs. The sample was coated with acrylic-based black matt paint to uniformise and enhance the surface emissivity, improving the effectiveness of PT inspection. The sample was placed on the optical table at a registered position within the working envelope of the robot arm, using a fixed custom supporting base. The specimen was inspected from the convex side. Figure 6c shows the sample ready for inspection. In order to validate the proposed alignment method, as will become clear in the following sections, the robotic thermographic inspection was also performed by wrapping the sample with a flexible plastic 3D printed grid (as shown in Figure 6d). The grid square pattern had a 3 mm pitch and wire width of 0.6 mm.

### 4.3. Robot Path-Planning, Simulation and Control

Six-axis robotic arms have traditionally been used in production lines to perform pick-and-place operations (e.g., palletising robots). In that scenario, where the exact trajectory between any two consecutive poses is not too important, a robot can be manually programmed by simply teaching the robot controller the coordinates of a few poses. Such teaching is usually performed by manually jogging the robot to each desired pose to record its coordinates. Then a robot programme is manually written to move the robot through the recorded poses. More recently, accurate mechanical joints and control units have made industrial robotic arms precise enough for finishing tasks in manufacturing operations [42]. As a result, software brands and robot manufacturers have developed many software applications to help technicians and engineers in programming complex robot tasks [43]. Using such software platforms to program robot movements is known as off-line programming (OLP). It is based on importing the 3D virtual model of the complete robot work cell, the robot end-effector, and the sample(s) to be manipulated or machined. Such robotic OLP software modules usually evolve from CAD/CAM applications, suited to programming Computer Numerical Control (CNC) manufacturing machines.

Despite the abundance of OLP software solutions geared towards manufacturing applications, limited solutions have been demonstrated for robotic NDT delivery [44,45]. Using commercial OLP software to generate appropriate tool paths for NDT purposes may seem relatively straightforward at first glance, but there are several inadequacies:Many commercial software applications for robotic off-line programming are expensive tools, incorporating a lot of functionality specific for CAD/CAM purposes and machining features;Path-planning for automated NDT inspections is a very particular task. Conventional OLP software has no accessible provision for tool-path customisation to accommodate the requirements of NDT inspections;Commercially available OLP software does not provide capabilities for full synchronisation between robotic movements and NDT data acquisition from sensor instrumentation systems (e.g., the thermographic IR camera, in this case). Such synchronisation is fundamental to enable the possibility of positional encoding of the NDT data to create accurate NDT maps of an inspected part [45].

In this work, robotic path-planning, simulation and control for automated thermographic inspection were enabled through developing a bespoke MATLAB-based graphic user interface. Figure 7 shows a screenshot of the application taken during the path-planning phase to inspect the sample described above. This software application imports the digital models of the robot, the thermographic instrumentation and the sample, producing a virtual representation of the inspection setup. The application was mainly developed to enable the automated thermographic data collection required for validating the data alignment method introduced by this work. Although it has no ambition to be a fully-developed software tool, it contains vital features to allow flexibility and future usability. The digital sample model is positioned in the virtual scene according to the user-specified coordinates for the sample reference frame with respect to the robot reference system. The set of coordinates comprises the three Cartesian coordinates of the sample origin and the three Eulerian angular coordinates of the coordinated axes. Although the application does not allow easy replacement of the employed thermographic instrumentation, provision has been made to enable customisation of the IR camera focal distance. Indeed, the inspection resolution depends on the camera’s distance from the sample surface for a given camera lens with a fixed focal distance. Thus, changing the camera focal distance is greatly important to allow accurate planning and simulation of the robotic task. The indication of the camera focal distance enables the software to compute the robot tool centre point (TCP) coordinates. The application allowed the creation of a raster inspection tool-path for the sample, according to the user-specified maximum spacing between consecutive image acquisition poses (25 mm) and offset from the sample edges (10 mm), resulting in an inspection path consisting of 15 data acquisition poses arranged in three passes (5 poses per pass). The TCP is kept on the part surface for all poses. The z-direction of the tool reference frame follows the surface’s normal direction to keep the camera view axis always perpendicular to the surface. Due to the curvature of the surface, the fact that the IR camera view axis is kept perpendicular to the surface does not guarantee that all the infrared rays emitted by the surface are perpendicular to the camera sensor. That aspect can be neglected by reducing the part surface area imaged from a single camera position, which is the main reason for employing robotic thermography. The surface area imaged from each camera position is reduced by bringing the camera closer to the part and/or cropping the camera’s full image frame (sub-windowing). The sub-windowing also allows higher frame acquisition rates, resulting in a better temporal sampling of the thermal wave.

The application allows simulating the automated task workflow before sending the path command coordinates to the connected robot. The connection between the computer and the robot was managed through the Interfacing Toolbox for Robotic Arms (ITRA) [46]. The ITRA allowed synchronising the robotic camera manipulation and the data collection to carry out the following steps, supported by the schematic representation given in Figure 5a:The computer sends the command coordinates of one inspection pose to the robot controller and waits for a digital acknowledgement from it, which signals the arrival of the robot arm at the commanded pose;While the robot is at a standstill, the flashlamp power supply is triggered;In turn, the sample surface temperature rise resulting from the flashlamp heat pulse triggers the IR camera data acquisition;The computer acquires the raw camera data through the FLIR ResearchIR Max^®^ software;The previous steps repeat for the following inspection pose until all poses are visited.

Figure 8 shows the robotic inspection system at the first five path poses. A video of the robotic data acquisition is available for download as Appendix A.

## 5. Results

Figure 9 and Figure 10 show the sets of thermographic images acquired with the tool path presented in Section 4.3, using a camera focal distance of 550 mm. The camera acquired the evolution of the thermographic field for 10 s at each pose (starting from one second before the trigger of the flashlamp). DFT was used to evaluate the frequency content of the thermal response at 0.6 Hz. All images have the same size (192 × 224 pixels). They relate to the images captured from the sample with and without the grid. Thus, the same robotic tool path was repeated twice to ensure repeatability in the acquisition poses. It must be noted how the average pixel intensity varies from image to image within each set; for example, image #4 and image #12 respectively present significant lower and higher average intensity than the rest of the images in the first set, respectively. Furthermore, pixel intensity is not repeatable since differences are evident across the two sets. Any pair of corresponding images in the two sets present a visible difference in pixel intensity.

Figure 11 highlights the initial estimate of the overlaps in each set of images. The images were encoded with the camera shooting positions and scaled by the measured resolution value. The known pitch of the grid (3 mm) was used to estimate the resolution of the images, which was 150 μm/pixel (≅4444 pixels/cm^2^). Figure 11a,b relate to the set of images with and without the grid, respectively. There, the image pixels were plotted with 50% transparency to allow visualising the overlaps, which are more clearly illustrated in Figure 11c. Following the notation introduced in Section 3.3, the presented FiPAM method was employed using a value of 1 mm for the offset (o) between the image overlap areas and the shrunk areas. This equates to assuming that the maximum distance between a pair of corresponding pixels in neighbour images (the misalignment) does not exceed 1 mm, which is the case for the sets of images at hand.

As stated above, the FiPAM algorithm contains the mathematical parametric formulation of all five typical 2D image transformations (translation, Euclidean, similarity, affinity and homography), allowing aligning all images in a set with the same type of planar transformation or using a different type of transformation for each image. Each image can be given a diverse set of DoFs and treated in six different ways if no transformation (no degrees of freedom) is included as an additional option, corresponding to a total of $6^{15}≅4.70\cdot 10^{11}$ possible diverse ways of applying FiPAM to our sets of fifteen images.

Figure 12 and Figure 13 illustrate the results obtained with FiPAM, using the similarity transformation for all images. The similarity transformation, which allows four DoFs (horizontal translation, vertical translation, rotation and scaling), proved sufficient to permit a fine alignment of all images in the given sets. In Figure 12a and Figure 13a, whereas the dotted blue line rectangles represent a fivefold scaled-down version of the original images, the rectangles with a green line perimeter represent a twofold scaled-down version of the aligned images, where the translation is magnified by a factor of 20 and the rotation transformation is maximised by a factor of 40. These magnifications were introduced to illustrate the computed alignment transformations for visualisation purposes. The infill colour given to each aligned image is linked to the computed pixel intensity correction through the indicated colour map. The resulting blended mosaic thermographic image (460 × 800 pixels) is given in Figure 12b and Figure 13b for the images with and without the grid, respectively.

Although testing FiPAM with all transformation combinations is not viable, the method was evaluated through a representative subset by employing each of the five possible planar transformations for all images and changing the number of images to align; this allowed varying the problem size significantly to evaluate the execution time of FiPAM. The number of images considered for each type of transformation was: 2, 5, 10 and 15, corresponding to aligning the first two images, the five images collected in the first pass of the tool path, the images in the first two passes or all the images in the set. As a result, the total number of DoFs considered in the alignment problem spanned from four, for two images transformed through pure translation (two parameters per image), to 120, for fifteen images transformed through homography (eight parameters per image). FiPAM was implemented and evaluated through MATLAB (version 2020b), running on a computer with an Intel^®^ i7-6820HQ CPU (2.70 GHz, 4 Cores) and 32 Gb of Random-Access Memory. The MATLAB implementation code developed in this work is accessible at https://doi.org/10.5281/zenodo.6817052 (accessed on 11 July 2022). The recorded execution times are plotted in Figure 14.

## 6. Discussion

Traditional manual inspection approaches are insufficient in some scenarios. Therefore, there are fundamental motivations for increasing automation in non-destructive testing. Automation of NDT is required to cope with the inspection of large and/or curved geometries. The key challenges to face when developing a robotic NDT system include integrating the NDT instrumentation with the robotic manipulator, creating a suitable robot inspection path for the part under inspection, and developing software for NDT data collection and visualisation. Although these challenges have been addressed by several applications of six-axis robotic arms for the inspection of parts through automated ultrasonic techniques, other types of inspections have not reached the same level of robotisation, which is the case with thermographic testing. This work bridges a technology gap, making thermographic inspections more deployable in industrial environments. Furthermore, the proposed fine image alignment method (FiPAM) can find applicability beyond thermographic NDT.

The results prove that FiPAM enables the proper merging of multiple thermographic images into one single mosaic image, which is easier to analyse. This is accomplished through three steps: simultaneous alignment of all images in a set, global optimum pixel intensity correction, and image blending. The reported composite mosaic images, in Figure 12b and Figure 13b, obtained through computing similarity transformations and pixel corrections for the images acquired in this work, show a significant reduction of the original discontinuities. Whereas the scale of the composite image relative to the sample with the grid is immediately retrievable from the known grid pitch (3 mm), a reference 20 mm long scale bar was added to the image relative to the sample without the grid. It is straightforward to note that the sizes of the thermographic indications correspond to the physical sizes of the artificial FBHs. The difference in thermographic pixel intensity for FBHs of diverse sizes is coherent with the change in the aspect ratio between heat blocking and leakage surface, as described in [47]. Larger FBH diameter to depth ratios produce the emergence of localised higher intensities in the IR image sequence.

Although FiPAM execution times are machine-dependent, the patterns presented in Figure 14 provide a helpful guideline for understanding the general trends. As expected, the execution times for alignment, pixel intensity correction and Laplacian blending increase with the number of images. The alignment phase execution time also depends on the type of transformations used for the images in the set. They influence the size of the least-squares problem and the number of transformation parameters to find through the minimisation of the SSD function in Equation (3). Thus, for a given number of images to align, using the same type of transformation for all images, the execution time increases monotonically, moving from translation to Euclidean, similarity, affinity and homography transformations. Although all possible combinations are not assessed in this work, it is not difficult to imagine intermediate execution times for generic combinations, where not all the images get transformed by the same transformation type. As a rule of thumb, for a given number of images, the alignment execution time should never exceed the time relative to the case where all images get transformed through homography, since it corresponds to the biggest problem with the maximum number of parameters. Fluctuations in alignment execution times can be observed for patterns relative to translation and Euclidean transformations. They are thought to be caused by the limited DoFs allowed by these transformations, which can cause prolonged convergence times due to the difficulty of obtaining a good image alignment. The execution times of the pixel correction and Laplacian blending phases depend on the number of images. The minor differences associated with the used transformation type are thought to be caused by the different overlaps of the aligned images, which changes the number of pixel intensity differences to compute for the SSD function in Equation (4).

The advantages of FiPAM, described in this work, should be clear by now. One limitation of the current implementation is that FiPAM is suitable for aligning multiple images of a sample surface that curves only in one direction. Although this limitation does not impede using FiPAM for generalised cylindrical surfaces, future work should focus on extending FiPAM to operate with images encoded in three-dimensional space.

## Figures and Tables

**Figure 1 sensors-22-06267-f001:**
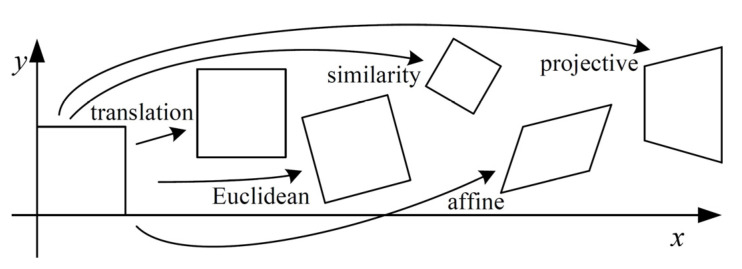
The basic set of 2D planar transformations (Reprinted with permission from Ref. [28]. 2007, now publishers inc).

**Figure 2 sensors-22-06267-f002:**
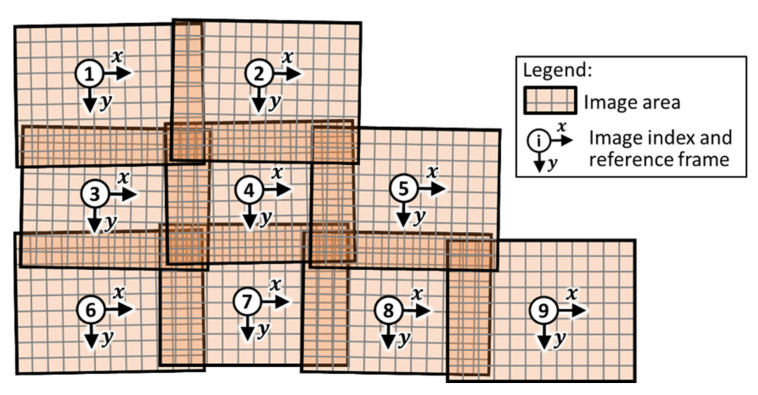
Schematic representation of a set of nine images used to explain the theoretical foundations of FiPAM.

**Figure 3 sensors-22-06267-f003:**
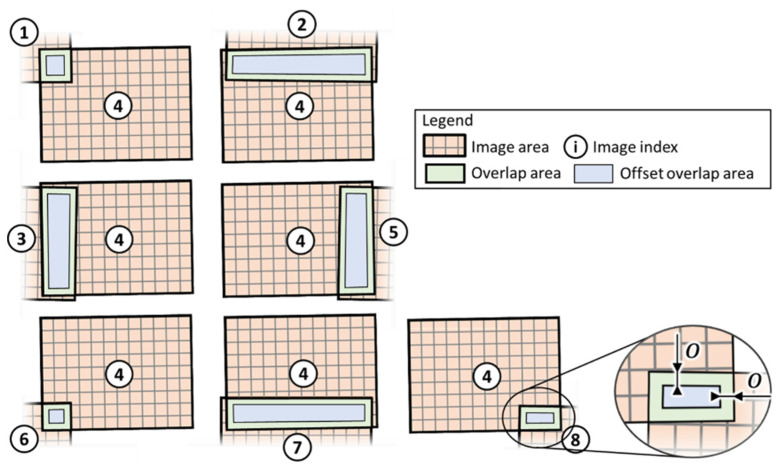
Illustration of all overlap areas between image #4 and neighbour images. The magnified region serves to clarify the relationship between the overlap areas and the offset overlap areas.

**Figure 4 sensors-22-06267-f004:**
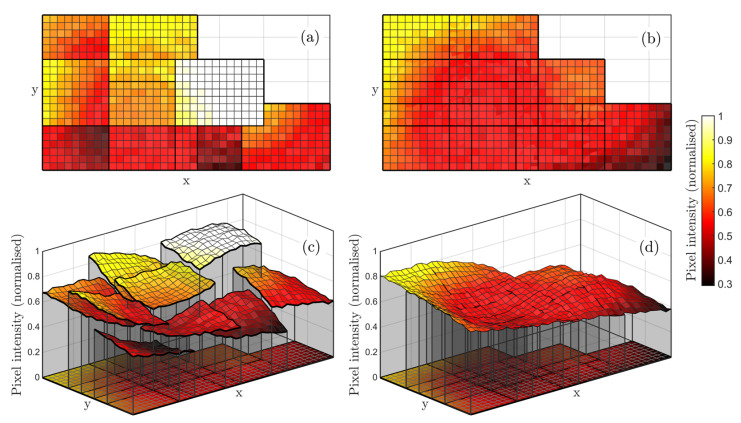
Exemplification of image blending, used in FiPAM. (**a**) Aligned images with discontinuous pixel intensities; (**b**) images after correction of discontinuities; (**c**) 3D plot of uncorrected pixel intensities; (**d**) 3D plot of the corrected image set.

**Figure 5 sensors-22-06267-f005:**
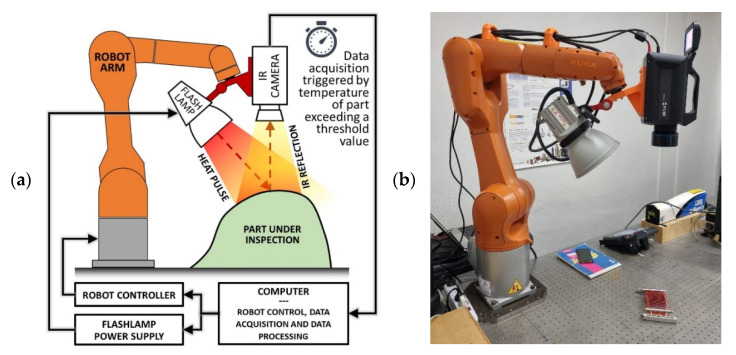
(**a**) Schematic representation of the robotic thermographic setup used in this work; (**b**) Photo of the actual laboratory setup.

**Figure 6 sensors-22-06267-f006:**
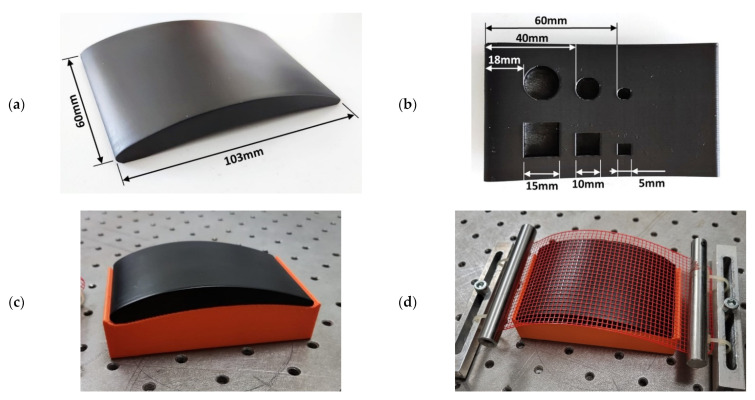
(**a**) Picture of the sample with the indication of footprint dimensions; (**b**) picture of the back wall with the indication of FBH locations and sizes; (**c**) sample placed on the supporting base without grid; (**d**) sample with the grid.

**Figure 7 sensors-22-06267-f007:**
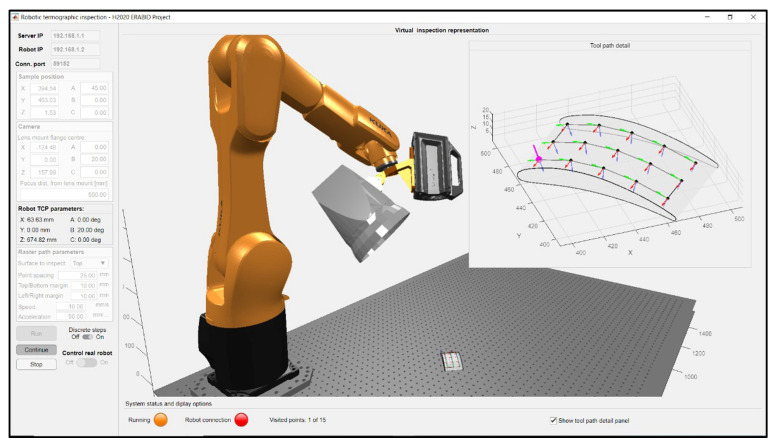
MATLAB-based graphic user interface for robot path-planning, simulation and control.

**Figure 8 sensors-22-06267-f008:**
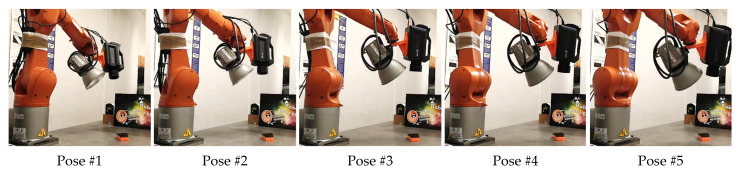
Robotic inspection system during data acquisition for the first five poses.

**Figure 9 sensors-22-06267-f009:**
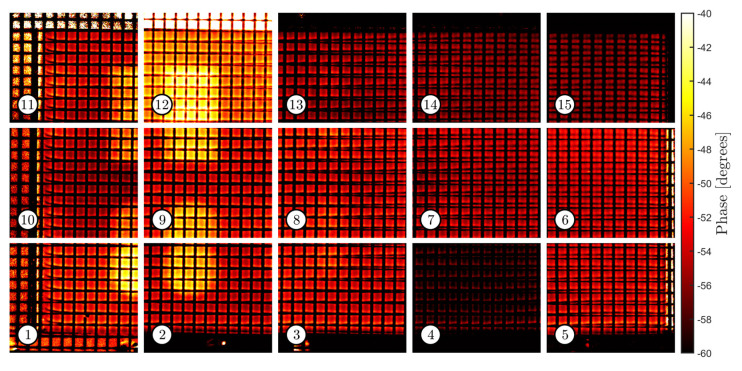
Set of phasegrams taken from the sample with the superposed grid.

**Figure 10 sensors-22-06267-f010:**
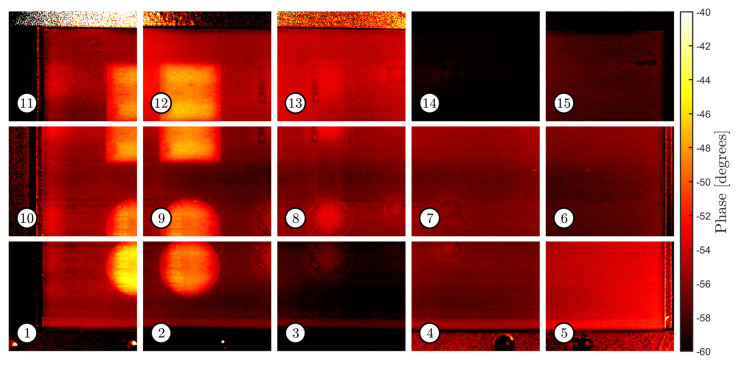
Set of phasegrams taken from the sample without the superposed grid.

**Figure 11 sensors-22-06267-f011:**
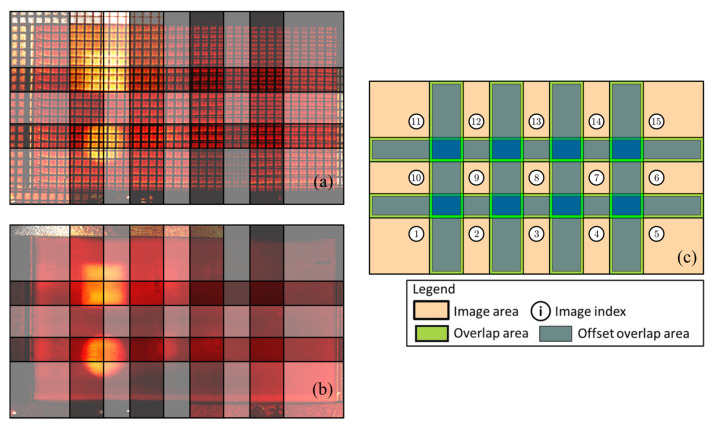
Plots of scaled and encoded images. (**a**) Set of images taken from the sample with the grid; (**b**) Images taken from the sample without the grid; (**c**) Overlapping areas.

**Figure 12 sensors-22-06267-f012:**
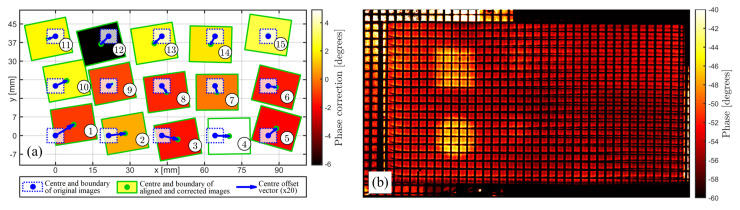
(**a**) Schematic illustration of similarity transformations and pixel intensity corrections computed through the proposed method for the set of images relative to the sample with the grid. (**b**) Resulting composite mosaic thermographic image.

**Figure 13 sensors-22-06267-f013:**
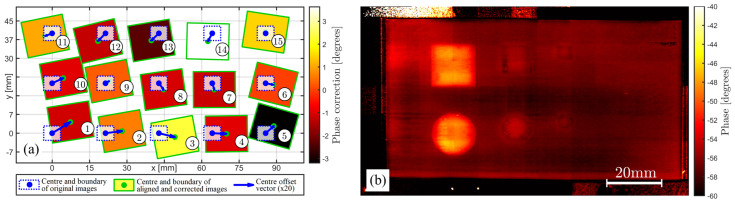
(**a**) Schematic illustration of similarity transformations and pixel intensity corrections computed through the proposed method for the set of images relative to the sample without the grid. (**b**) Resulting composite mosaic thermographic image.

**Figure 14 sensors-22-06267-f014:**
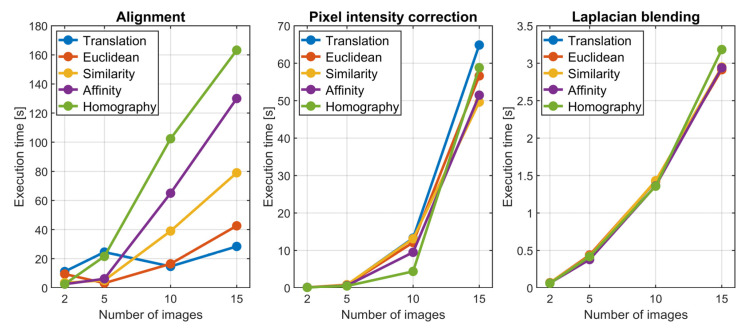
Execution times for alignment, pixel intensity correction and Laplacian blending.

## Data Availability

The source code for the MATLAB-based implementation of FiPAM and the example thermographic dataset are available at https://doi.org/10.5281/zenodo.6817052 (accessed on 11 July 2022).

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
