# Peer review of "Fine Alignment of Thermographic Images for Robotic Inspection of Parts with Complex Geometries"

_sensors, 2022, doi:10.3390/s22166267_

Round 1

Reviewer 1 Report

Authors should review and consider these papers and discuss how their approach differs from or adds to this prior works. These references do not represent and exhaustive research in a area of much active research.

Holland, S.D., Krishnamurthy, A. (2022). Registration of NDE Data to CAD. In: Meyendorf, N., Ida, N., Singh, R., Vrana, J. (eds) Handbook of Nondestructive Evaluation 4.0. Springer, Cham. https://doi.org/10.1007/978-3-030-73206-6_5

Xianlin Meng, Yonghui Wang, Junyan Liu, Wantao He, Nondestructive inspection of curved clad composites with subsurface defects by combination active thermography and three-dimensional (3D) structural optical imaging, Infrared Physics & Technology, Volume 97,2019,Pages 424-431,ISSN 1350-4495,

https://doi.org/10.1016/j.infrared.2019.01.026

Combining modern 3D reconstruction and thermal imaging: generation of large-scale 3D thermograms in real-time, Schramm, Sebastian and Osterhold, Phil and Schmoll, Robert and Kroll, Andreas, Quantitative InfraRed Thermography Journal},pages=1--17,2021

https://doi.org/10.1080/17686733.2021.1991746

Comments on technical content of paper:

Section 2. Thermography Principles

Line 129 : “Once raw data is collected, the Discrete Fourier Transform (DFT) is calculated to evaluate the frequency content of the thermal response.”

This is one of the ways the data can be analyzed, however, there are multiple techniques that have been shown to be equally effective. This line should be included in 4.1. Inspection System Integration

3.2. Fine Pixel-based Alignment Method

Starting at line 237: “It must be noted that FiPAM is currently suitable for aligning multiple mosaic images of a sample surface that curves only in one direction. Although the constrain of single direction curvature is a significant limitation, that does not impede using FiPAM for mosaic images of any surface belonging to the large family of cylindrical surfaces intended as "generalized cylindrical surfaces" [32].”

This would seem to require more specifications as to the limits of the technique. For a small diameter cylinder the mapping of pixel area to surface area of the cylinder changes along the radius of the cylinder. Therefore, the perspective transformation in equation 1 would not seem to be accurate. What degree of curvature is allowed in the specimen for this technique to be applicable.

3.3. Image Blending

The phase shifts in the phasegrams shown in figure 4 seem very large relative to what is typically seen in phasegrams. An advantage of the generating the phasegram is the effects of variations in the intensity of the excitation across the surface are significantly reduced. Is this artificial data? Also, what is the process used to register the phasegrams based on the intensity of the images as is discussed in the section “3.2. Fine Pixel-based Alignment Method” if there are these large discontinuities at the edges of neighboring images.

4.1. Inspection System Integration

 Line 357 “Error! Reference source not found. illustrates the automatic thermography setup used in this work.“

Typical of Word reference changing during editing.

5. Results

What is the frequency of the phasegrams? What is the duration of the data acquisition?

In figure 10, there are dramatic differences in phase between some of the neighboring images such as 13 and 14 where the change is about 8 degrees. Was the raw data examined to understand the reason for this change? Before Fourier transforming the data, how is the data during and immediately after the flash handled? Does preprocessing the time records before performing the Fourier Transform result in reduced phase changes from segment to segment. In general, more information is needed on how the phasegrams images were formed. 

Reviewer 2 Report

The article titled "Fine Alignment of Thermographic Images for Robotic Inspection of Parts with Complex Geometry" is focused on the thermographic investigation of defects in parts for industrial production.

The authors claim to have introduced a new "insepction platform" that consists of 3 main "components": 1. a pulsed thermography system, 2. a six-axis robotic manipulator, 3. an algorithm for image alignment, correction and blending.
In particular, at line 92 the authors stress the novelty introduced in the algorithm for the image processing (the other 2 components of the system appear quite standard, as stated by the authors themselves at lines 15 and 16).

The introduction describes the problem to be faced. In the lines between 62 and 68 a short comparison with ultrasound techniques (topic on which some of the authors worked in the past, see citations) is reported. Other techniques for NDT are anyway available but have not been cited. This appears more as an expedient to introduce self citations than a way to provide the reader a complete description of the state of the art. I suggest to reduce the number of not useful self citations and shortly describe other techniques for NDT with relative references.
At line 76 the authors refer to the "reflection mode": it would be fine to give a short sentence to state how the reflection mode works.
At line 82 the expression "optimal positions" appears. The word "optimal" is used in the scientific literature to design the best condition among the available ones. Anyway the description on how to achieve this optimality is missing in the paper.

In sec 3.2 a Mathematical summary and formalization of the geometric deformations available for the images matching are provided.
The used notation is not particularly clear: the authors work in 2D and they define x to be the vector of coordinates (2 coordinates). Then, e.g. in (1), they multiply the 2-dimensional x by a 3 by 3 matrix: a clarification is needed.

Moreover, at line 185 s is defined as "parameters": is s a scalar? Is the  scaling operation isotropic?

The notation used in the formalization affects also the subsequent definition of the SSD. In general, the SSD is a functional by which minimization the registration occurs. This kind of functionals have been widely used in the image processing literature, also with different aims (e.g., inpainting) but references are totally missing.

The introduced novelty reduces then to the simultaneous registration of multiple images. 
The formalization of this peculiarity of the algorithm is given in (3) (affected by the problems concerning the notation exposed before). 
The advantage of the simultaneous minimization is clear nor from a numerical point of view (analysis of complexity is missing) nor from a computational one (a heuristic to prove a possible computational improvement for the convergence of the functional is not given). A comparison between the minimization of images taken by pair and the simultaneous case is opportune.

At line 270, to reduce the computational time in the minimization of the SSD, a numerical value for the offset is provided without any exaplanation on how to choose it.

The authors underline that it is possible to choose the suitable transformation for each of the overlapping images but no description of the criteria used to operate this choice is given: this is far away from an automatization of the NDT, that is the goal of the proposed method.

At line 292 the need for interpolation is stated and the bicubic approximation is chosen: many interpolation techniques are available and no references are provided to support the best performances of the bicubic.

In Section 3.3, at the beginning, the blending is explicitly avoided because values close to the measurements are needed (lines 315, ,316) but then it is introduced at the end of the procedure to have smooth transitions: these behaviour seems contradictory at a first glance. A deeper explanation of the reasons is opportune.

By the conducted experiment the technique has been tested on a 10 cm long part. Maybe the adjective large used in the article is not the most appropriate to refer to parts of this dimensions.

Same for the shape of the used experimental sample that not appears so complex. 

For the above reasons I suggest a major revision of the paper, before considering the publication.

Minor issues:

Using a more frequent indentation can improve the readability of the manuscript;

I would prefer the abbreviation Eq. in places of Equ.;

end of line 72: I suggest to add "or a scene and it is based...";

line 129: "raw data are collected";

line 221: "can not"

line 254: I suggest to use the word "reference" in place of "template"

line 289: "follows" in place of "descends"

fig. 4, put the figure in the same order used in fig. 6 (a, b top; c, d down);

In sec. 4 a reference is missing (line 357);

a flowchart could be useful to describe the actions taken from line 479 to line 487;

line 496: "image #4 and image #12 respectively present..."

biography: generally it seems a standard format is missing

In [3] the authors names are repeated.

A list of all the used abbreviations could improve the readability of the article

Round 2

Reviewer 2 Report

The requested corrections have been included.